# Academic achievement of students without special educational needs and disabilities in inclusive education–Does the type of inclusion matter?

Grzegorz Szumski[1]*, Joanna Smogorzewska[1], Paweł Grygiel[2]

**1** Department of Education, University of Warsaw, Warsaw, Poland, **2** Jagiellonian University, Institute of Education, Cracow, Poland

* g.szumski2@uw.edu.pl

**Data Availability Statement:** The data are available from the Dryad Digital Repository: Szumski, Grzegorz; Smogorzewska, Joanna; Grygiel, Paweł

## Abstract

The main aim of the study, conducted in Poland, was to compare the academic achievement of 1552 (at Time 1) students without disabilities in three educational settings: general, without students with disabilities, inclusive (with co-teaching), with three to five students with disabilities, and inclusive (without co-teaching), with one to two students with disabilities. The study was longitudinal, with three waves eight months apart. The latent growth curve model was used for data analysis. The results have shown that changes in academic achievement (for language and for mathematics) over time were similar in all three groups; therefore, there were no differences between inclusive education (of the two types) and general education classrooms. These results mean that students neither lose nor benefit while learning in inclusive education classrooms. The findings are in line with the results of previous meta-analyses and are important for the future development of inclusive education.

## Introduction

Although ethical and legal considerations have been the main engine for the development of inclusive education, they need to be supported by empirical evidence [1]. Consider, for example, a hypothetical result that demonstrates educational benefits for students without disabilities taught in inclusive classrooms. Such a finding would be a strong argument for further implementation of inclusive education, as it shows that it responds to the needs of all students [2]. In contrast, empirical evidence that an inclusive classroom is a less conducive environment than a traditional classroom for the progress of the students without disabilities would challenge those who promote inclusive classrooms [3]. It would also call into question the concept of inclusive education as a transformational model for education systems [4].

One of the key outcomes studied is academic achievement, which is analysed more often than social development, mental well-being, or quality of life at school [5,6].

This study provides the first analysis of the academic achievement of students without disabilities using a latent growth curve model, which allows for comparing the achievement of

(2022), "Academic achievement of students without special educational needs and disabilities in inclusive education," PlosOne, Dryad, Dataset, https://doi.org/10.5061/dryad.j3tx95xhj.

**Funding:** The study was financed by National Centre for Science (NCN) in Poland (no. 2012/07/B/HS6/01434). The funders had no role in study design, data collection and analysis, decision to publish, or preparation of the manuscript.

**Competing interests:** The authors have declared that no competing interests exist.

students in inclusive and traditional classrooms the changes in this achievement, and also in tracking the trajectories of such changes. Moreover, we analyse the effects of both forms of inclusive education, when used concurrently in one country, a topic that has rarely been analysed [7]. Comparison of the effectiveness of these two forms is important for choosing a strategy for further promoting inclusive education. Last, but not least, it is the first study about the academic achievement of students without disabilities in inclusive classrooms conducted in a Central Eastern European region. It has been shown in a recent meta-analysis that a context built by the school systems and educational policies of individual countries and regions is meaningful for the academic achievement of students without disabilities in inclusive classrooms [3].

## The development of inclusive education

The term 'inclusive education' is subject to many interpretations. One of the latest attempts to classify various definitions distinguishes four main ways in which inclusive education is understood in the literature [2]. The common feature in the use of 'inclusive education' is the placement of students with disabilities in general education classrooms [8]. This perspective has its roots in a fight against the systematic exclusion of students with disabilities by placing them in special education classrooms. However, the lifting of barriers to access to regular schools and general education classrooms is, on its own, insufficient to provide a high-quality education and good socio-emotional experiences for students with disabilities [9]. This observation brought motivation to clarify and narrow the notion of inclusive education by complementing it with indicators of quality, such as high academic achievement, positive relations with typically developing peers, and participation in school activities [10,11].

Although placing students with disabilities in general education classrooms changes the environment of education for typically developing students, this aspect was at first rather marginalised in debates about inclusive education. However, with time, the reflection about inclusive education was widened to the whole class, or even the entire school. From this holistic thinking arose the concept of inclusive education as one of high quality for all students [12]. Even though this is increasingly widely accepted in academic discourse and policy, it can lead to confusion due to contradictory ideas about the characteristics of a good school, and, more importantly, what values it promotes. Some educational academics and politicians state that an inclusive school should, in the first place, create conditions for students to maximise their potential and achievement, including those who were previously often marginalised and even excluded from the system [13–15]. It thus fits into a neoliberal model of education, which is based on acknowledging economic rationality and on limited understanding of social justice to only the rule of equal opportunities, rejecting the role of schools in the process of building a community [16,17]. Most people think, however, that inclusive education is a vision of schools standing *against* a neoliberal model, or even trying to eliminate its limitations [18,19]. Adherents of this community-based approach support the maximisation not of individual achievement, but rather of community, which a school that prepares children to live in society.

In the current work, we adopt an understanding of inclusive education as a school good for all students, although the scope of our research is limited to academic achievement of students without disabilities. However, because we want to check whether inclusive education fosters high achievement of students without disabilities, we also use a wider understanding of inclusion, thus placing students with disabilities in mainstream classrooms. This procedure is recommended by researchers, who advise a precise differentiation of inclusive education as a method from effects of its practising [2,11].

When the promotion of inclusive education becomes a long-term aim of educational policies [20], the implementation of inclusion in schools must be as effective as possible. It requires research to examine the effects of different forms of inclusive education [21]; this may be interventional research that tests the effects of new solutions at a classroom, school or school district level, or naturalistic research evaluating different forms of inclusive education [22]. This study is an example of the latter type of research.

## Academic achievement of students without disabilities in inclusive classrooms, and its determinants

Most research on the effectiveness of inclusive education for typically developing students concerns academic achievement [3,6]. A meta-analysis of 47 studies from six countries has shown that the presence of students with disabilities in primary and secondary classrooms has a positive, but weak, impact on the achievement of their peers without disabilities [3]. Strong or moderate effects were rarely observed [23,24]. Additionally, the importance of a few moderators was established. A positive effect was visible in studies conducted in the USA, but not in Europe. It also occurred in elementary education but was not present at higher levels. In addition, it was acknowledged that the type and level of the students' disability were relevant moderators. A few extensive studies from Europe using system data, published after this meta-analysis, compensate for a shortage of studies from this region [25–27]. In European educational systems, students without disabilities do not benefit from learning in inclusive classrooms, but they also do not lose out in comparison to those in traditional classrooms. Only one Finnish study produced more ambiguous results–when students without disabilities in inclusive and traditional classrooms were compared, no differences in achievement were observed at the end of the first level in secondary school, despite a higher initial level of achievement by students in traditional classrooms [25].

The weak effects observed in most studies, and an indication that the design of a study and the way of analysing the data can distort the effect of inclusive classrooms on the achievement of students without disabilites, are strong motivation for designing new studies in this area [3,28]. These new studies should be longitudinal and test the key moderators and mediators of the effects.

## Co-teaching and alternative models of inclusive education

Although inclusive education may seem quite homogeneous, many models of implementation exist [9,29]. These solutions are sometimes specific to each country, but they also vary within countries [25,30]. Solutions are determined by various factors, be it the kinds of disabilities of students and the level of their need for additional support, the schools' resources, the competencies of class teachers and special education teachers, policies concerning diagnoses of disabilities, and the placement of students with disabilities [31]. One of the key criteria for categorising forms of inclusive education is a special support provided [32]. Students may receive support in a classroom from special education teachers or paraprofessionals, sometimes complemented with extra support outside the classroom; or they may only receive additional help outside the classroom, which in most cases is quite limited. When the class teacher teaches simultaneously with a special education teacher it is called 'co-teaching', practised in many countries worldwide [9]. Most researchers differentiate co-teaching from employing paraprofessionals to support students with disabilities in inclusive classrooms [9]. However, sometimes this distinction fades. Paraprofessionals often have training for working with students with disabilities [30], and the tasks they perform are not different from those of special education teachers in a co-teaching model [33]. In at least some European countries, including

Austria, Germany, Switzerland and Poland, co-teaching is combined with a reduction in the number of students in the classroom and with regulations concerning the ratio of students with and without disabilities [34–36]. In these and other countries where students with disabilities are included in 'typical' classrooms, they receive additional support for a few hours per week outside the classroom [34,36,37].

## Effectiveness of co-teaching–assumptions and empirical evidence

In a co-teaching system, students with disabilities can be grouped in the inclusive classroom and at the same time provided with intensive, additional support. There is also the possibility that students with disabilities are dispersed among different classrooms and provided only with extensive personal resources. Such differences affect not only the quality of education for their typically developing peers, but also the idea of inclusive education as a transformation of the school system toward schooling which is good for all [38]. The extra resources in the form of an additional teacher in the classroom can be seen as protection against lower quality teaching for typically developing students, and as a solution for the improvement of academic achievement in comparison to traditional school classrooms.

Special education teachers or qualified teachers' assistants (TAs) protect class teachers from overload resulting from teaching students with disabilities. However, they are rarely treated as equal teachers in the classroom. In most cases, they adjust he teaching for students with disabilities and help them during the lesson or organise alternative teaching, while the class teacher instructs the whole class [39]. Such a solution gives class teachers more opportunity for interaction with typically developing students than in classrooms where only one teacher is working [40].

The positive effect of teaching students without disabilities in inclusive classrooms can, however, be decreased by disruptive behaviour by students with disabilities [41]. Therefore, teachers are more sceptical about the inclusion of students with emotional and behavioural disorders than those with other kinds of disabilities [42]. Special education teachers or TAs in inclusive classrooms are often responsible for undertaking *ad hoc* interventions with students displaying disruptive behaviour, by, for example, asking them to calm down or leaving the classroom with the student to talk to him/her. But in co-teaching it is easier to use proactive solutions to prevent such behaviours [33,43,44]. A recent observational study found that co-teaching reduces the number of incidences of disruptive behaviour by students with disabilities [45].

The belief that inclusive education can improve the educational outcomes of all students is mostly driven by an expectation of using proven, high-quality teaching strategies [9]. Additional resources, such as a second teacher or qualified paraprofessional in the classroom can provide elements of individualisation of the teaching, such as giving students frequent relevant feedback, adapting materials and teaching methods to the individual needs of the student, and using adequate methods to track progress [46]. However, teaching strategies in co-teaching classrooms are in most cases no different to those used in inclusive classrooms where only one teacher works, or in traditional classrooms without students with disabilities [33,45,47]. There are three reasons why co-teaching may not increase the quality of teaching. Firstly, relations between class teachers and special education teachers are often not based on partnership, the latter being treated only as helpers [39]. Secondly, class teachers and special education teachers lack time for planning lessons together, which makes it difficult to share ideas or coordinate cooperation [48–50]. Thirdly, special education teachers often lack the subject knowledge to be able to teach together with class teachers [51]. This is especially the case in higher level classes, as the content of teaching is more advanced [5]. So, while there are some good examples of

co-teaching, the general picture is mixed, making hypothesising about its effectiveness difficult.

## Socioeconomic status and access to inclusive education

Together with spreading inclusive education, studies on inequalities in access to education and schools providing high-quality education for students with and without disabilities are important. There are many social factors, such as the socioeconomic status (SES) of parents, the cultural capital of the family, race, or place of residence, which can affect the educational opportunities of students. Proportionally more intersectional studies have been conducted concerning students with disabilities [52,53]. Results of these studies show, first of all, that a lower parental SES and belonging to a racial minority increase the probability of the occurrence of disability in children [54,55]. Moreover, students from such environments often have few opportunities to learn in inclusive classrooms [56,57]. However, fewer studies focus on the factors to take into account when placing students without disabilities in inclusive classrooms; this is important for the practical aspect of the concept of inclusive education, understood as a good school for all students. It is known that the composition of schools and classrooms is important for the progress of students due to the 'peer effect' [58,59], and also because of worse outcomes in classrooms with an overall lower SES [60]. Overrepresentation of students with low social and cultural capital brings risks to making the idea of inclusive education real. Although the belief is often repeated that inclusive education classrooms often accept students with low SES [9], there is a shortage of strong empirical proof for this; there is a lack of random group studies that directly relate to this issue. As a matter of fact, only one study, conducted in England, directly indicates a tendency to group, in the same classroom, students with and without disabilities coming from families with low SES [61]. Canadian research shows a significant relations between the number of students with disabilities and the SES of students; but in this case, the SES was assessed indirectly, as that of the inhabitants of the school district rather than that of parents of the students in the classroom [62]. In contrast, a huge study conducted in the US, also at the level of school district, has shown that a higher level of inclusive education spread is visible in districts where fewer students get free/reduced-price lunch and is not connected with the salaries of the children's parents [52]. The hierarchical regression analyses have indicated that none of the measures of students' family SES predict the level of inclusive education spread [52]. Therefore, because of rare and inconsistent results, there is a need for further studies concerning the relationship between parental SES and inclusive education. The current study seeks to fill this gap.

## Inclusive education in Poland

Inclusive education in Poland has been intensively promoted since the early 1990s, the beginning of democratic changes. During the school year 2016/2017, 56% of students with disabilities were taught in inclusive education classrooms, but at the lower level of secondary school it was a little lower than this [63]. Promotion of inclusive education was possible thanks to (1) special education reforms, such as introducing a common core curriculum for typically developing students and most students with disabilities, improving funding for special education in inclusive classrooms (changes in educational law, 1998, 1999, see: Żyta et al., 2017) [64], and (2) ratifying the United Nations Convention on the Rights of Persons with Disabilities [35,63]. Parents of children with disabilities also play an important role, and they can choose between regular and special schools [65].

   Three decades ago, two types of inclusive placement were shaped in Poland, and continue in use today [39]. One is 'inclusive' classroom, in which co-teaching is used and the number of

students is reduced (to 20), with a maximum of three to five students with disabilities. Co-teachers have qualifications in special education, so they are not paraprofessionals. In addition, students with disabilities learning in these classrooms have access to therapeutic support outside the classroom for a few hours per week, due to their diagnosis of needs and Individual Educational Programmes. The second kind of placement is a regular class in which individual students with disabilities learn, called individual inclusion. In this type of setting, general education teachers give instruction and adapt programmes for the needs of students with disabilities. Although these students get additional, special support outside the classroom, it is not a significant amount–around two to four hours per week [35,65].

The two types of placement are promoted unevenly–approximately twice as many students with disabilities learn in classes without co-teaching than in classes with co-teaching [35]. However, in big cities where we conducted the study, the number of co-teaching classes is higher than the country's mean and students with disabilities have access to both types of placement.

## The current study

The aim of the study was to analyse and compare the academic achievement of students without disabilities learning in three types of classroom in Poland. We elaborated this general aim into three separate research questions:

1. Do students from traditional and inclusive classrooms exhibit differences in the level of their family SES and initial academic achievement?

2. Does the academic achievement of students without disabilities in language (Polish) and in mathematics change over time differently in inclusive classrooms in comparison to traditional classrooms?

3. Does the academic achievement of students without disabilities in language (Polish) and in mathematics change over time differently in inclusive classrooms with co-teaching or without?

## Methods

### Participants

In the study, 1813 native Polish students without disabilities took part in the longitudinal study. In the initial Time 1 (T1) study, 1552 students took part, in T2, 1521 students, and in T3, 1483 students participated. 1171 students (64.6%) participated in all three waves, 491 students (22.1%) took part in two waves, and 241 students (13.3%) participated in one wave.

We selected first-grade lower secondary (the 7th year of education) students based on multi-level and multi-strata criteria. During conducting the study, which took place just before the last educational reform in Poland (2017), the educational system had three stages, which were in accordance with the ISCED classification [65,66]: elementary (K-5), lower secondary (6–8), and upper secondary (9–12) education. Each educational stage means starting a new school. The sample was drawn using the Polish Educational Information System (PEIS, https://cie.men.gov.pl/sio-strona-glowna). Classes were selected randomly (excluding special education classes and classes in small schools with fewer than 10 students per grade). We invited two or three randomly chosen first-grade classes from each school to participate in the study.

Students learned in 108 classrooms. 55 (50.9%) were general education classrooms in which 58.2% of participants learned; 23 (21.3%) of classrooms were inclusive without co-teaching in which 22.6% of participants learned; 30 (27.8%) of classrooms were inclusive with co-teaching, in which 19.2% of all students learned. The mean number of students in each classroom was $M = 16.79$ ($SE = 0.6$; $Min = 3$; $Max = 29$).

During the first study there were altogether 174 students with statements of disabilities in inclusive classrooms, whose results are not analysed in the current study. The most frequently occurring disabilities were mild intellectual disability ($n = 45$, of which $n = 40$ in co-teaching classrooms), physical disabilities ($n = 28$, of which $n = 25$ in co-teaching classrooms), autism spectrum disorder ($n = 27$, of which $n = 25$ in co-teaching classrooms), and deaf and hard of hearing ($n = 18$, of which $n = 14$ in co-teaching classrooms).

Teachers of Polish language and mathematics, working in three types of classrooms, did not express any differences in work experience.

The sample's detailed demographic data are presented in Table 1.

## Measures

**Achievement tests.** Students were asked to solve three achievement tests both in mathematics and in Polish language. In the case of mathematics, the test consisted of 16 tasks, which were in line with the curriculum. The tasks were prepared in accordance with the curriculum. The reliability of the tests was appropriate: $\alpha = 0.72$ (wave 1), $\alpha = 0.82$ (wave 2), and $\alpha = 0.79$ (wave 3). There were six tasks used as anchor tasks, repeated in each wave, as well as nine tasks repeated in two waves (one and two, or two and three). In the case of language tests, in wave one there were 14 tasks, and in waves two and three there were 16 tasks in each test. The reliability of the tests was acceptable: $\alpha = 0.71$ (wave 1), $\alpha = 0.75$ (wave 2), and $\alpha = 0.79$ (wave 3). Six tasks were used as anchor tasks in all three waves, and nine tasks repeated in two waves (one and two, or two and three). Students received one point for each correct answer in the tests. The types of tasks in language and mathematics tests in the three waves are presented in Table 2.

**Socioeconomic status (SES).** We created the short index of the SES, based on a principal component analysis of four variables: (1) the mean level of education of the mother and the father, (2) the mean self-assessment of living conditions; (3) the self-assessment of the level of life satisfaction; (4) the number of books in the household. These variables were responsible for almost half (49.6%) of the variance in SES.

## Procedure

Students were asked to fill in the questionnaires in language and mathematics a few days apart in their classrooms, as a group. The procedure was repeated three times approximately eight months apart. Written consent was obtained from school principals and classroom teachers, as well as from students' parents, and students also agreed verbally to participate. The research

**Table 1. Demographic data.**

|  | General education classes | Inclusive classes without co-teaching | Inclusive classes with co-teaching | Together |
|---|---|---|---|---|
| **Students characteristic** | | | | |
| Age (*M*, *SD*) | 13.01 (.44) | 13.09 (.53) | 13.07 (.53) | 13.04 (.48) |
| Gender (% of girls) | 49.1 | 52.3 | 47.7 | 49.6, mean proportion of boys to girls: $M = 1.18$ (range 0.06–3.75), $F(2, 100) = 0.92$, $p =$ ns. |
| **Teachers characteristic** | | | | |
| Work experience–Polish language (*M*, *SD*) | 16.67 (9.34) | 18.00 (9.15) | 17.07 (10.11) | 17.06 (9.45), $F(2, 96) = 0.138$, $p =$ ns. |
| Work experience–mathematics (*M*, *SD*) | 17.48 (8.00) | 20.06 (7.91) | 17.42 (7.73) | 17.95 (7.88), $F(2, 83) = 0.705$, $p =$ ns. |

**Table 2. Types of tasks across the three tests in Polish language and mathematics.**

| Type of skill | Test 1 –T1 (number of tasks) | Test 2 –T2 (number of tasks) | Test 3 –T3 (number of tasks) |
|---|---|---|---|
| **Polish language** | | | |
| Grammar | 5 | 7 | 5 |
| Vocabulary | 5 | 5 | 5 |
| Orthography & punctuation | 2 | 2 | 4 |
| Text comprehension | 1 | 1 | 1 |
| Writing short statements | 1 | 1 | 1 |
| **Mathematics** | | | |
| Word problems (including spatial geometry, scale, fractions, algebra, calendar calculations) | 7 | 4 | 5 |
| Algebra | 1 | 1 | 2 |
| Geometry | 5 | 4 | 4 |
| Fractions | 2 | 5 | 3 |
| Figures & tables comprehension and interpretation | 1 | 2 | 2 |

project was positively assessed by an Ethical Committee for Scientific Research at the University of Warsaw, Department of Education (no.1/2018).

## Data analysis and missing data

The results of our study are presented in the following order. Firstly we present descriptive statistics and correlations of the gathered data. Then we present growth curve models (three time points) of the results for mathematics and language in three educational settings, preceded by information about the adjustment of the model to the data as well as information on the invariance of results in the waves. Two growth curve models are presented: one without co-variants, and the other with two co-variants: gender and SES.

Missing data were handled with FIML estimation. In FIML, the parameters (and standard errors) of a statistical model are estimated using a likelihood function for each participant based on observed relationships between non-missing data, without imputing or removing critical information from the data set [67].

## Results

### Preliminary analysis

Descriptive statistics and correlations between variables are presented in Table 3 (without co-variants) and in Table 4 (with co-variants).

The results show that tests in mathematics were slightly more difficult for students than the language tests, but all show moderately high achievement. Correlations between results in language and mathematics within the same wave, as well as with those in subsequent waves, are rather strong.

### The main analysis

Students learning in traditional classrooms had a higher level of family SES ($M = 0.047$) than respondents from inclusive classrooms without co-teaching ($M = -0.03$) or with co-teaching ($M = -0.10$): $F(2, 1810) = 4.25$, $p = 0.014$, $\eta^2 = 0.005$. Results concerning the initial academic achievement of the students in the three groups are described in the growth curve models below. However, the results show no significant differences among the groups.

**Table 3. Means, standard deviations, and correlations with confidence intervals (without covariants).**

| Variable | M | SD | 1 | 2 | 3 | 4 | 5 |
|---|---|---|---|---|---|---|---|
| 1. Mat–T1 | 0.49 | 0.21 | | | | | |
| 2. Mat–T2 | 0.48 | 0.24 | .58** | | | | |
| | | | [.54, .61] | | | | |
| 3. Mat–T3 | 0.43 | 0.23 | .53** | .60** | | | |
| | | | [.49, .57] | [.56, .63] | | | |
| 4. Lang–T1 | 0.58 | 0.25 | .58** | .51** | .46** | | |
| | | | [.55, .62] | [.47, .55] | [.42, .50] | | |
| 5. Lang–T2 | 0.61 | 0.25 | .56** | .65** | .49** | .63** | |
| | | | [.53, .60] | [.62, .67] | [.45, .53] | [.60, .66] | |
| 6. Lang–T3 | 0.57 | 0.26 | .50** | .52** | .63** | .60** | .60** |
| | | | [.46, .54] | [.48, .56] | [.60, .66] | [.56, .64] | [.56, .63] |

*Note*. M and SD are used to represent mean and standard deviation, respectively. Lang–results in language (three waves), Mat–results in mathematics (three waves). Values in square brackets indicate the 95% confidence interval for each correlation. The confidence interval is a plausible range of population correlations that could have caused the sample correlation (Cumming, 2014).

* indicates $p < .05$.

** indicates $p < .01$.

**Mathematics achievement.** Table 5 presents fit indices for results in mathematics for three waves and three groups.

The results showed that prepared models fit the data well, so further steps could be taken. We tested only between groups, only longitudinal, as well as between groups and longitudinal invariance. The analysis of invariance indicated that on each wave the model reached the scalar

**Table 4. Means, standard deviations, and correlations with confidence intervals (with covariants).**

| Variable | M | SD | 1 | 2 | 3 | 4 | 5 | 6 | 7 |
|---|---|---|---|---|---|---|---|---|---|
| 1. Lang–T1 | 0.58 | 0.25 | | | | | | | |
| 2. Lang–T2 | 0.61 | 0.25 | .63** | | | | | | |
| | | | [.60, .66] | | | | | | |
| 3. Lang–T3 | 0.57 | 0.26 | .60** | .60** | | | | | |
| | | | [.56, .64] | [.56, .63] | | | | | |
| 4. Mat–T1 | 0.49 | 0.21 | .58** | .56** | .50** | | | | |
| | | | [.55, .62] | [.53, .60] | [.46, .54] | | | | |
| 5. Mat–T2 | 0.48 | 0.24 | .51** | .65** | .52** | .58** | | | |
| | | | [.47, .55] | [.62, .67] | [.48, .56] | [.54, .61] | | | |
| 6. Mat–T3 | 0.43 | 0.23 | .46** | .49** | .63** | .53** | .60** | | |
| | | | [.42, .50] | [.45, .53] | [.60, .66] | [.49, .57] | [.56, .63] | | |
| 7. Gender (Girls) | 0.50 | 0.50 | .23** | .25** | .23** | .09** | .06* | .01 | |
| | | | [.19, .28] | [.20, .29] | [.18, .28] | [.04, .14] | [.01, .11] | [-.04, .06] | |
| 8. SES | 0.00 | 0.88 | .24** | .29** | .25** | .27** | .31** | .28** | .03 |
| | | | [.19, .29] | [.24, .33] | [.20, .30] | [.22, .32] | [.26, .35] | [.23, .33] | [-.02, .07] |

*Note*. M and SD are used to represent mean and standard deviation, respectively. Lang–results in language (three waves), Mat–results in mathematics (three waves), SES–socioeconomic status. Values in square brackets indicate the 95% confidence interval for each correlation. The confidence interval is a plausible range of population correlations that could have caused the sample correlation (Cumming, 2014).

* indicates $p < .05$.

** indicates $p < .01$.

**Table 5. Fit indices for results in mathematics.**

| Model | N | No. Param. | $\chi^2$ | | | RMSEA | | | CFI | TLI |
|---|---|---|---|---|---|---|---|---|---|---|
| | | | $\chi^2$ | df | p | RMSEA | 90CI LB | 90CI UB | | |
| Time I (All) | 1551 | 32 | 162.5 | 104 | .000 | .019 | .013 | .025 | .949 | .941 |
| Time I (General) | 902 | 32 | 117.6 | 104 | .172 | .012 | .000 | .022 | .981 | .978 |
| Time I (Inclusive without co-teaching) | 356 | 32 | 122.0 | 104 | .109 | .022 | .000 | .037 | .913 | .900 |
| Time I (Inclusive with co-teaching) | 293 | 32 | 128.5 | 104 | .052 | .028 | .000 | .043 | .857 | .835 |
| Time II (All) | 1521 | 32 | 158.0 | 104 | .001 | .018 | .012 | .024 | .976 | .972 |
| Time II (General) | 872 | 32 | 129.1 | 104 | .048 | .017 | .002 | .025 | .981 | .978 |
| Time II (Inclusive without co-teaching) | 355 | 32 | 115.1 | 104 | .216 | .017 | .000 | .033 | .976 | .973 |
| Time II (Inclusive with co-teaching) | 294 | 32 | 12.6 | 104 | .127 | .023 | .000 | .040 | .965 | .960 |
| Time III (All) | 1476 | 34 | 204.1 | 119 | .000 | .022 | .017 | .027 | .960 | .954 |
| Time III (General) | 871 | 34 | 16.3 | 119 | .007 | .020 | .011 | .027 | .968 | .964 |
| Time III (Inclusive with co-teaching) | 319 | 34 | 126.4 | 119 | .304 | .014 | .000 | .032 | .975 | .972 |
| Time III (Inclusive with co-teaching) | 286 | 34 | 143.3 | 119 | .064 | .027 | .000 | .041 | .939 | .930 |

invariance, after releasing one loading and one threshold of one task in T2 and T3 in the group learning in the classrooms with co-teaching, allowing for comparisons between groups (Scalar-$^{MOD1}$: $\Delta\chi2 = 126.67(102)$, $\Delta p = .049$, RMSEA = .008, CFI = .966, TLI = .964; Scalar$^{MOD2}$: $\Delta\chi^2 = 123.51(101)$, $\Delta p = .064$, RMSEA = .008, CFI = .966, TLI = .965). This scalar-invariant model was used as a base for modelling latent changes in results.

The latent growth curve model without co-variants was well-fitted to the data (RMSEA = .008, CFI = .964, TLI = .963). The intercepts in all groups are not significantly different from zero, the slopes (growth of results over time) were different for each group. Students from general education classrooms significantly improved their results with time (Est = 0.19, SE = 0.04, $p = .0001$). Results of students from both forms of inclusive setting did not change significantly over time (Est = 0.052, SE = 0.106, $p = .622$ & Est = 0.103, SE = 0.103, SE = 0.056, $p = .065$, respectively). The comparison of groups' changes in results did not bring any significant results, and the growth of results in mathematics did not differ between groups; general education versus inclusive non-co-teach classrooms: Est = 0.137, SE = 0.113, $p = .225$; general education versus co-teach classrooms: Est = 0.087, SE = 0.066, $p = .190$; inclusive non-co-teach classrooms versus co-teach classrooms: Est = -0.051, SE = 0.119, $p = .672$. The model with co-variants was also well-fitted to the data (RMSEA = .009, CFI = .960, TLI = .958). In the case of the group of students in general education classrooms it turned out that adding gender as a covariant did not have any impact on the intercept (Est = 0.120, SE = 0.075, $p = .107$) or on the slope (Est = - 0.041, SE = 0.050, $p = .412$) of mathematics achievement. SES as a covariant had impact on the intercept in this group of students (Est = 0.260, SE = 0.069, $p = .0001$), but not on the slope (Est = 0.030, SE = 0.031, $p = .337$). In the case of students in inclusive classrooms without co-teaching, gender as a covariant was not significant for the intercept of the results in mathematics (Est = 0.159, SE = 0.097, $p = .103$), but was significant for the slope (Est = -0.164, SE = 0.08, $p = .05$), showing that among boys the growth of the results in mathematics was higher than among girls. SES, in turn, was significant for the intercept (Est = 0.387, SE = 0.112, $p = .001$), as well as for the slope (Est = -0.129, SE = 0.04, $p = .002$), indicating that students from lower SES homes made better progress than students from higher SES homes. In the group of students learning in classrooms with co-teaching, the gender of students was significant not only for the intercept (Est = 0.168, SE = 0.084, $p = .046$), but also for the slope (Est = -0.149, SE = 0.074, $p = .043$), showing again that boys made significantly better progress than girls. SES, however, impacted the intercept (Est = 0.205, SE = 0.062, $p = .001$), but not the slope

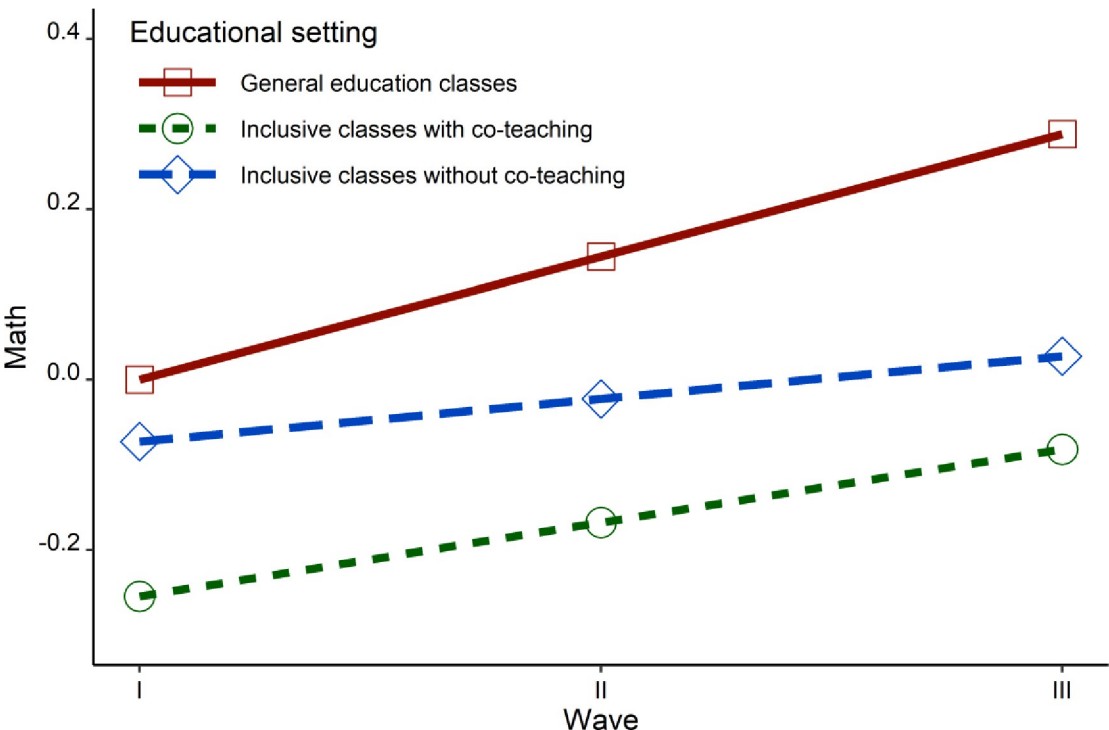

**Fig 1. Changes in academic achievement in mathematics across waves and educational settings.**

(Est = 0.059, SE = 0.033, *p* = .069) of the results. There were, however, no significant differences between groups in the case of changes in the results over time; general education classrooms versus inclusive classrooms without co-teaching: Est = 0.102, SE = 0.093, *p* = .272, general education classrooms versus co-teaching classrooms: Est = 0.086, SE = 0.060, *p* = .148, inclusive classrooms without co-teaching versus co-teaching classrooms: Est = -0.016, SE = 0.099, *p* = 0.868. Fig 1 presents changes in academic achievement in mathematics across waves and educational settings.

**Language achievement.** Table 6 presents fit indices for results in language for three waves and three groups.

The results showed again that the prepared models suit the data well, so further steps could be taken.

We, therefore, tested only between groups and only longitudinal, as well as between groups and longitudinal invariance. The analysis of invariance indicated that on each wave the model reached the scalar invariance, after releasing one loading and one threshold of one task in T3 in the group learning in the classrooms with co-teaching, allowing for comparisons between groups (Scalar$^{MOD1}$: $\Delta\chi2$ = 133.42(108), $\Delta p$ = .050, RMSEA = .008, CFI = .980, TLI = .979). This scalar-invariant model was used as a base for modelling latent changes in results.

The latent growth curve model without co-variants fit the data well (RMSEA = .008, CFI = .979, TLI = .978). The intercepts in all groups are not significantly different from zero, the slopes (growth of results over time) were significant in each group. Students from all groups significantly improved their results over time (general education classrooms: Est = 0.255, SE = 0.045, *p* = .0001; inclusive classrooms without co-teaching: Est = 0.149, SE = 0.056, *p* = .008; inclusive classrooms with co-teaching classrooms: Est = 0.184, SE = 0.053, *p* = .0001). However, the comparisons of the changes in groups' results did not show any significance, so

**Table 6. Fit indices for results in language.**

| Model | N | No. Param. | χ² | | | RMSEA | | | CFI | TLI |
|---|---|---|---|---|---|---|---|---|---|---|
| | | | $\chi^2$ | df | p | RMSEA | 90CI LB | 90CI UB | | |
| Time I (All) | 1552 | 28 | 83.0 | 65 | .065 | .013 | .000 | .021 | .990 | .988 |
| Time I (General) | 901 | 28 | 84.9 | 65 | .049 | .018 | .001 | .029 | .980 | .976 |
| Time I (Inclusive without co-teaching) | 357 | 28 | 71.0 | 65 | .285 | .016 | .000 | .036 | .985 | .982 |
| Time I (Inclusive with co-teaching) | 294 | 28 | 71.7 | 65 | .265 | .019 | .000 | .041 | .981 | .977 |
| Time II (All) | 1521 | 32 | 177.6 | 90 | .000 | .025 | .020 | .031 | .962 | .956 |
| Time II (General) | 873 | 32 | 141.2 | 90 | .001 | .026 | .017 | .033 | .955 | .948 |
| Time II (Inclusive without co-teaching) | 353 | 32 | 100.8 | 90 | .205 | .018 | .000 | .035 | .983 | .980 |
| Time II (Inclusive with co-teaching) | 295 | 32 | 88.8 | 90 | .515 | .000 | .000 | .031 | 1.000 | 1.000 |
| Time III (All) | 1483 | 34 | 129.2 | 104 | .048 | .013 | .001 | .019 | .989 | .988 |
| Time III (General) | 874 | 34 | 123.9 | 104 | .089 | .015 | .000 | .024 | .984 | .982 |
| Time III (Inclusive without co-teaching) | 321 | 34 | 106.6 | 104 | .412 | .009 | .000 | .031 | .994 | .993 |
| Time III (Inclusive with co-teaching) | 288 | 34 | 106.5 | 104 | .413 | .009 | .000 | .032 | .994 | .993 |

the growth of results did not differ between groups: general education versus inclusive classrooms without co-teaching: Est = 0.106, SE = 0.062, $p$ = .088; general education versus classrooms with co-teaching: Est = 0.071, SE = 0.059, $p$ = .226; inclusive classrooms without co-teaching versus those with co-teaching: Est = -0.035, SE = 0.072, $p$ = .628.

The model with co-variants also fit the data well (RMSEA = .0008, CFI = .968, TLI = .967). In the case of a group of students in general education classrooms it turned out that adding gender as a covariant had an impact on the intercept (Est = 0.366, SE = 0.081, $p$ = .0001), but not on the slope (Est = - 0.011, SE = 0.042, $p$ = .796) of language achievement. SES as a covariant had an impact on the intercept in this group of students (Est = 0.200, SE = 0.045, $p$ = .0001), but not on the slope (Est = 0.032, SE = 0.025, $p$ = .192). In the case of students in classrooms without co-teaching, gender as a covariant had an impact on the intercept of the results in language (Est = 0.429, SE = 0.125, $p$ = .001), but was not significant for the slope (Est = 0.063, SE = 0.053, $p$ = .23). SES, in turn, was significant for the intercept (Est = 0.260, SE = 0.068, $p$ = .0001), as well as marginally for the slope (Est = 0.069, SE = 0.036, $p$ = .058), indicating that students from higher SES homes made better progress than students from lower SES homes. In the group of students learning in classrooms with co-teaching, the gender of the students was significant for the intercept (Est = 0.309, SE = 0.101, $p$ = .002), but not for the slope (Est = 0.04, SE = 0.066, $p$ = 0.546). SES, however, impacted the intercept (Est = 0.217, SE = 0.056, $p$ = .0001), but not the slope (Est = -0.007, SE = 0.032, $p$ = .821) of the results in language. There were, again, no differences in the improvement of the results between groups; general education classrooms versus inclusive classrooms without co-teaching: Est = 0.087, SE = 0.058, $p$ = .134, general education classrooms versus classrooms with co-teaching: Est = 0.074, SE = 0.060, $p$ = .216, inclusive classrooms without co-teaching versus classrooms with co-teaching: Est = -0.013, SE = 0.069, $p$ = .084. Fig 2 presents changes in academic achievement in language across waves and educational settings.

## Discussion

This study had three main aims. Firstly, we checked, with a longitudinal sample of lower level secondary students, representative of Polish big cities, whether schools use a hidden selection of students without disabilities for inclusive classrooms on account of initial academic achievement and family SES. Secondly, we analysed whether achievement in language and

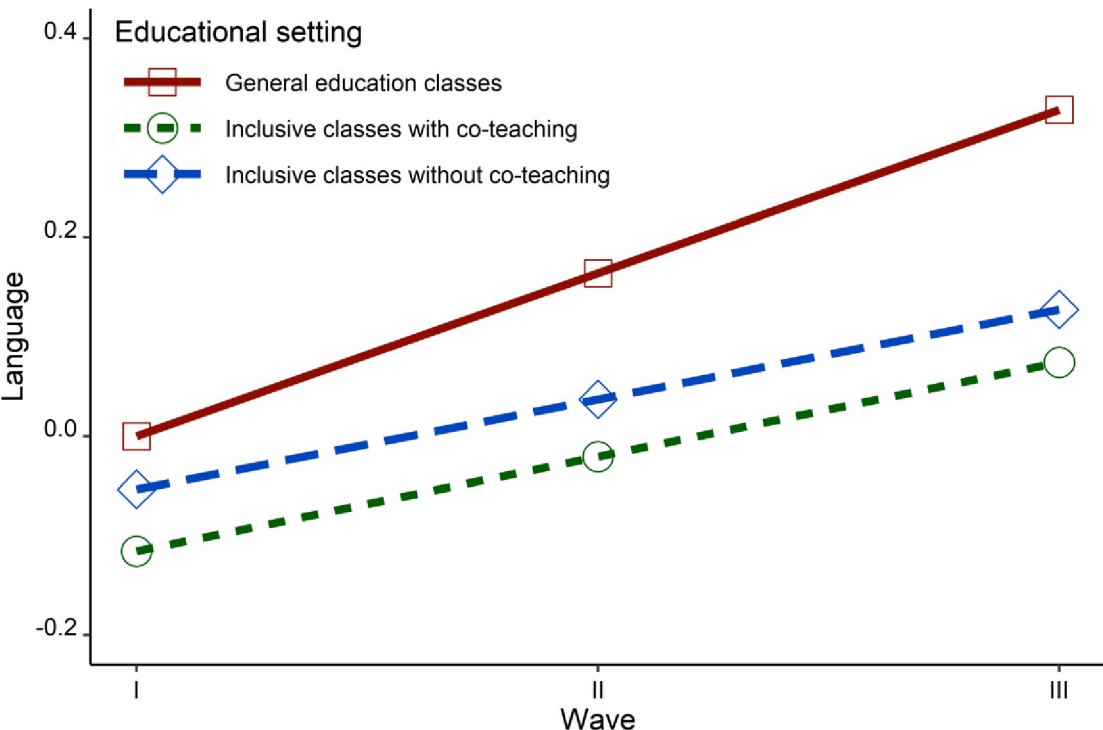

**Fig 2. Changes in academic achievement in language across waves and educational settings.**

mathematics of typically developing students in inclusive classrooms changes faster than the achievement of their peers in traditional classrooms. Thirdly, we observed whether academic achievement in language and mathematics of the students in inclusive classrooms with co-teaching changes faster than the achievement of their peers in inclusive classrooms without additional support. Although the two latter questions were crucial for this study, we will start our analyses from the selection of students without disabilities for inclusive classrooms.

Our study has shown that in the Polish educational system there are differences between types of educational settings on account of family SES, but not on account of the initial achievement of students. Students learning in traditional classrooms had a higher level of family SES than respondents from inclusive classrooms without co-teaching or classrooms with co-teaching. Importantly, in those schools where we conducted our research, there were three types of educational settings, and parents were responsible for the final decision concerning their child's placement. The mechanisms for composing each type of inclusive classroom in Poland are different. A school cannot place students without disabilities in classrooms with co-teaching without parental permission, but at the same time, schools freely set up inclusive classrooms without a special education teacher. Despite different regulations for composing school classrooms, in both cases, hidden selection on the basis of SES was observed, showing that this phenomenon is universal. Such a tendency is also visible in other countries [25,61,62], including Finland, known for caring about a policy of 'equality and quality' in education, which should be associated with a lack of tracking [68].

In summary, the results of our study may indicate that not only many teachers, but also many parents with high SES, perceive inclusive classrooms as a risk factor for the progress of those students without disabilities. Such beliefs conflict with the concept that inclusive education is a way to improve schooling, including improving the quality of education, and can

threaten the realisation of this idea. It is because lower SES predicts lower achievement, not only on an individual level, but also on a group level. The academic achievement of students in classrooms with, on average, a lower SES, is lower than in classrooms with higher SES, after controlling for factors on an individual level [59]. Various factors can be responsible for this result. For instance, because students with lower SES and with disabilities have a higher tendency to have behavioural problems, an inclusive classroom can be disadvantageous for the improvement of academic achievement [69]. Another potential danger can be employing less experienced and less effective teachers in inclusive classrooms. International comparative studies show that better qualified and more experienced teachers more often teach in classrooms in which students come from families with higher SES and in which there are no students with disabilities [60]. This mechanism has not only an inter- but also an intra-school character.

The second issue that is interesting for us was a comparison of the increase in academic achievement in language and mathematics in inclusive and traditional classrooms, with two forms of inclusive education (with and without co-teaching). The results have shown that changes in the academic achievement of students without disabilities in both types of inclusive education do not differ from changes observed in traditional classrooms, neither in the case of language nor mathematics. In other words, typically developing students neither lose nor benefit while learning with peers with disabilities. This result is in accordance with the majority of previous research. Importantly, it agrees with the results from a meta-analysis of studies of mixed designs, which showed that typically developing students in European school systems do not benefit from learning in inclusive classrooms [3]. It is also in line with the results of the latest meta-analysis, which showed that according to longitudinal studies, the inclusion in classrooms of students with generalised difficulties in learning had no significant impact on the academic progress of their peers without such difficulties [70]. It is also important to notice that this meta-analysis showed a significant difference in the extent of the effect in longitudinal and cross-sectional studies–the latter showed a weak negative effect of inclusive education on the academic achievement of typically developing students [70]. Although the relationship between study design and results is difficult to explain, it shows a need to conduct more longitudinal research, as it is more reliable for the assessment of the importance of inclusive education for the academic achievement of typically developing students.

It is also important to notice that even though we did not observe differences in increase between inclusive and traditional settings, the mathematical abilities of students grew only in traditional classrooms. A weak increase in mathematical abilities of students in the lower level of secondary education is not surprising, because it was already observed in previous research [71]. However, our study observed a lack of students' progress in mathematics only in inclusive classrooms, raising concerns about the effectiveness of this educational setting, already mentioned in previous analyses [72].

The third issue raised in our study was a comparison of the growth of academic achievement of students learning in two forms of inclusive education–in a co-teaching model and in a model without additional support in the classroom. In this analysis, we did not observe any significant differences between the changes in the academic achievement of students, neither in the Polish language nor in mathematics. While discussing this result we need to, first of all, remember that the settings compared differed not only in the presence or absence of a special education teacher in the classroom, but also in the number of students with disabilities present. In co-taught classrooms, there were more students with disabilities than in classrooms without co-teaching. Our results, therefore, bring more knowledge about popular and specific placement than about the effectiveness of co-teaching *per se*. However, it shows that co-teaching does not create a new quality of education, bringing a better learning and teaching

environment, at least in naturalistic conditions. Co-teaching, therefore, does not imply such a change of teaching methods in the classroom that it fosters improvement in the academic achievement of students without disabilities. At best, it protects from a possible decrease in their achievement in classrooms where there is a high percentage of students with disabilities, who need adaptations of the programmes and additional support. The fact that additional resources protect from such a decrease was already noticed in large studies conducted in other countries [27]. Our result is in contrast with the result of Murawski and Swanson's meta-analysis [73], a quantitative research review about co-teaching, as well as with later studies [47].

It is, however, important to consider the huge differences between our study and those mentioned before. Firstly, studies considered by Murawski and Swanson, as well as the study by McDuffie et al., were interventions, while our study concerned typical educational practices. Interventions are in principle better prepared and bring more satisfactory results than every-day educational practices. Secondly, most of these studies concerned preschool and elementary school children, and not secondary school students. Previous research points out that at lower education levels inclusive education is more effective than among older students [3,70]. Thirdly, in many studies included in the meta-analysis of Murawski and Swanson [73], grades were used as a measure of achievement, which can bring different results from using the results of achievement tests.

## Practical implications

The results of our study have practical implications. First of all, they show that it is possible to include individual students with disabilities in regular classrooms with limited special support. Such a solution does not decrease the academic achievement of students without disabilities. Although such a model is in part contrary to the contemporary conception of inclusion that focuses on the role of multi-specialist teams in providing for the needs of all students, it should not be abandoned. In many countries and regions, where access to special education teachers is limited, such a model of inclusion may be the only method of providing fairly rapid access to school for students with disabilities [74].

## Limitations and directions for future research

Although in the present study a fairly strong longitudinal design with repeated measures was used, together with a recommended method of data analysis–the latent growth curve model– the study has a few limitations. Firstly, as in most previous research on this topic, we did not control for the cognitive abilities of the participants. Secondly, we did not observe the teaching methods used by teachers in individual classrooms. Such observations could be helpful in explaining the results obtained. Unfortunately, a large number of schools did not allow us to make such observations. Thirdly, the span of the research was rather short and did not allow for following the changes in academic achievement during the whole period of learning in lower level secondary schools, let alone the further academic careers of the students. In addition, with the ability to include more time points, we may have been able to have a clearer idea regarding what happens to achievement in the longer term. Fourthly, no post hoc power analyses were performed since the only available population parameter estimates (needed for such an estimation) are based on the sample at hand, and consequently, post hoc power analysis would not be informative with regard to the interpretation of results [75]. These elements should be added in future analyses to obtain even more valuable results [76].

## Author Contributions

**Conceptualization:** Grzegorz Szumski.

**Data curation:** Grzegorz Szumski, Paweł Grygiel.

**Formal analysis:** Joanna Smogorzewska, Paweł Grygiel.

**Funding acquisition:** Grzegorz Szumski.

**Methodology:** Grzegorz Szumski, Joanna Smogorzewska.

**Project administration:** Grzegorz Szumski.

**Visualization:** Paweł Grygiel.

**Writing – original draft:** Grzegorz Szumski, Joanna Smogorzewska.

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
