## [Decision Letter · Decision Letter 0]

14 Feb 2022

PONE-D-21-30610Academic achievement of students without special educational needs and disabilities in inclusive education - does the type of inclusion matter?PLOS ONE

Dear Dr. Szumski,

Thank you for submitting your manuscript to PLOS ONE. I have now heard from knowledgeable reviewers and upon my own reading of the paper and after careful consideration, we feel that it has merit but does not fully meet PLOS ONE’s publication criteria as it currently stands. Therefore, we invite you to submit a revised version of the manuscript that addresses the points raised during the review process.

The referees and I see value in this paper, given the novelty that it brings to a scarce literature on the effect of inclusive education on the achievement of students without special needs or disabilities.

Given the clear expertise of the referees on this topic, I defer to their comments and will ask you to simply respond to them. However, my view is that the main areas of concerns, which you would do well to address in rewriting the paper, are as follows:

1) add a short description of the Education System in Poland (R2)

2) explain, from a theoretical point of view, which concept of inclusive education are you referring to  (R1)

3) revise thoroughly the whole manuscript to improve readability (R3)

Beyond the above highlighted points, as I noted previously the referee reports are all of high quality, so please make sure to respond directly to individual referee comments.

Congratulations on the work so far; I look forward to reading the revision.

We look forward to receiving your revised manuscript.

Kind regards,

Simona Lorena Comi

Academic Editor

PLOS ONE

https://journals.plos.org/plosone/s/fileid=ba62/PLOSOne_formatting_sample_title_authors_affiliations.pdf".

“GS - UMO- 2012/07/B/HS6/01434 - Narodowe Centrum Nauki, Poland, www.ncn.gov.pl, funder does not play any role in the study or the article preparation”

“The study has been financed by National Centre for Science (no. UMO- 2012/07/B/HS6/01434), Poland”We note that you have provided funding information that is not currently declared in your Funding Statement. However, funding information should not appear in the Acknowledgments section or other areas of your manuscript. We will only publish funding information present in the Funding Statement section of the online submission form.

“GS - UMO- 2012/07/B/HS6/01434 - Narodowe Centrum Nauki, Poland, www.ncn.gov.pl, funder does not play any role in the study or the article preparation”

Reviewers' comments:

Reviewer's Responses to Questions

**Comments to the Author**

1. Is the manuscript technically sound, and do the data support the conclusions?

Reviewer #1: Yes

Reviewer #2: Yes

2. Has the statistical analysis been performed appropriately and rigorously? 

Reviewer #1: Yes

Reviewer #2: I Don't Know

3. Have the authors made all data underlying the findings in their manuscript fully available?

Reviewer #1: Yes

Reviewer #2: Yes

4. Is the manuscript presented in an intelligible fashion and written in standard English?

Reviewer #1: Yes

Reviewer #2: No

5. Review Comments to the Author

Reviewer #1: The paper aimed to explore the academic achievement of students without special educational needs and disabilities (SEND) in inclusive classrooms. Actually, it is the first research conducted in a Central Eastern European region and the first analysis of the academic achievement of students without SEND using a latent growth curve model.

On the one hand, I found the paper to be overall well written and much of it to be well described. I felt confident that the authors performed careful and thorough latent growth curve model processing. On the other hand, I found some of the description of the paper to be taken into consideration as an ad-hoc fact without any references.

I explain my concerns in more detail below. I ask that the authors specifically address each of my comments in their response.

Major comments:

- In the section “The development of inclusive education” you mention the research approach to be adopted in order to understand inclusive education, but since an enormous underlying problem with inclusive education relates to lack of agreement about what constitutes an effective inclusive school environment, I would suggest to provide the readers with a definition that you comply with.

See: Van Mieghem, A., Verschueren, K., Petry, K., & Struyf, E. (2020). An analysis of research on inclusive education: a systematic search and meta review. International Journal of Inclusive Education, 24(6), 675-689.

- In the section “Measures”, please provide the readers with a Table which will describe the three achievement tests in mathematics and in Polish and their tasks.

- In the section “Limitations and directions for future research”, please mention also that you have not conduct a Post-hoc power analysis and explain shortly why not.

- In the section “Limitations and directions for future research”, you can also add the following phrase: “In addition, with the ability to include more time points, we might have been able to get a clearer idea regarding what happens to achievement in the longer term”.

Minor comments:

- The manuscript includes grammatical and syntax corrections. Please, provide a final version.

- Page 2, Line 4: the acronym/abbreviation T1 has to be explained.

- Page 2, Line 16: keyword students without special educational needs to be replaced by students without special educational needs and disabilities

- Page 4, 11: needs a different word than ambiguous. I would suggest: “Inclusion in education is subject to many interpretations”

- Page 5, Line 20: Add a short explanation (two extra lines maximum with a citation) of the neoliberal model of education.

- Page 6, Line 14: How many of these 47 studies concerned students from primary and secondary education? Please clarify.

- Page 10, Line 18: Citation 53, add more recent citation to reinforce the argument such as:

a. Sirkko, R., Takala, M., & Wickman, K. (2018). Co-teaching in northern rural Finnish schools. Education in the North, 25(1–2), 217–238.

b. Scruggs, T. E., & Mastropieri, M. A. (2017). Making inclusion work with co-teaching. Teaching Exceptional Children, 49(4), 284–293.

- Page 11, Line 5: Please mention the specific special education reforms in parenthesis.

- Page 11, Lines 8-9: “Parents of children with SEND … rather than special schools”: this is a strong statement please provide readers with a citation.

- Page 11, Lines 11-20: Please refer to the law/reform that describes this framework.

- Page 11, Lines 23-25: Please provide readers with citation of the law which justifies the higher number of co-teaching classes.

All best wishes,

Reviewer #2: Thank you for the opportunity to review the manuscript, “Academic achievement of students without special educational needs and disabilities in inclusive education - does the type of inclusion matter?”. The field is in need of more research on this topic, and I think this paper will be an important addition to the literature. Below I list my concerns with the current version of the manuscript:

Abstract-Specify that the study was conducted in Poland.

Introduction- Please consider reorganizing your literature review to create a more cohesive flow. There are some short paragraphs that could be combined and long ones that can be split. Also, I suggest moving the paragraph on page 3 line 21 to the current study subsection. Similarly, the first paragraph on page 6 and the second paragraph on page 7 should be moved to the current study subsection for a better flow. Additionally, I think the authors need to explain Poland’s educational system in terms of the educational stages and grades (could be in the inclusive education in Poland subsection). For example, on page 12 authors stated that they selected first-grade students. What is the grade equivalent of this in terms of PreK to 12?

• Page 9 line 25 please explain what ad hoc interventions are. The reader needs examples to understand what this means.

Method- consider adding a demographics table to the participant’s subsection. Also, the authors stated that there were 174 students with disabilities in inclusive classrooms. It would be interesting to see their academic growth trajectories. If this was not planned as another paper, please consider broadening the aim of the paper for also students with disabilities and add a separate analysis.

Results- no suggestion

Discussion- The authors said the study had three main aims, but the first aim was not mentioned in the paper earlier. I suggest putting this information in the introduction and rewriting the research questions. Page 20 line 5 “Our study has shown that in the Polish educational system there is informal tracking of…..” I am unconvinced that the study finds informal tracking. Please explain more. Page 20 line 18 Authors said, “In summary, the results of our study indicate that not only many teachers, but also many well-educated parents perceive inclusive classrooms as a risk factor for the development of students without SEND.” Please explain how your study indicates this conclusion.

Terminology- It's not clear why the authors use students without special educational needs and disabilities instead of only students without disabilities.

I believe the authors should revise the overall manuscript to improve the readability of the text.

Some minor issues

• Page 7 line 23 “This diverse solutions…..” consider revising the sentence

• Page 8 line 22 “For instance, ……..” consider revising the sentence

• Page 9 line 13 “Most studies show that……” please provide a reference

• Page 9 line 20 “The positive effect” consider starting a new paragraph

6. PLOS authors have the option to publish the peer review history of their article (what does this mean?). If published, this will include your full peer review and any attached files.

Reviewer #1: **Yes: **Dr. Sofia Mastrokoukou, Research Fellow at the University of Turin, Department of Pscychology

Reviewer #2: No

---

## [Author Response · Author response to Decision Letter 0]

27 Apr 2022

Reviewer #1: The paper aimed to explore the academic achievement of students without special educational needs and disabilities (SEND) in inclusive classrooms. Actually, it is the first research conducted in a Central Eastern European region and the first analysis of the academic achievement of students without SEND using a latent growth curve model.

On the one hand, I found the paper to be overall well written and much of it to be well described. I felt confident that the authors performed careful and thorough latent growth curve model processing. On the other hand, I found some of the description of the paper to be taken into consideration as an ad-hoc fact without any references.

I explain my concerns in more detail below. I ask that the authors specifically address each of my comments in their response.

Major comments:

!) In the section “The development of inclusive education” you mention the research approach to be adopted in order to understand inclusive education, but since an enormous underlying problem with inclusive education relates to lack of agreement about what constitutes an effective inclusive school environment, I would suggest to provide the readers with a definition that you comply with.

See: Van Mieghem, A., Verschueren, K., Petry, K., & Struyf, E. (2020). An analysis of research on inclusive education: a systematic search and meta review. International Journal of Inclusive Education, 24(6), 675-689.

A1: Thank you very much for your suggestion. We made changes in this section and we made use of the paper you mentioned. 

2) In the section “Measures”, please provide the readers with a Table which will describe the three achievement tests in mathematics and in Polish and their tasks. 

A2: We added a table describing types of tasks used in the tests (Table 2).

3) In the section “Limitations and directions for future research”, please mention also that you have not conduct a Post-hoc power analysis and explain shortly why not. 

A3: We added this limitation and we explained the reasons why we could not conduct this analysis. 

4) In the section “Limitations and directions for future research”, you can also add the following phrase: “In addition, with the ability to include more time points, we might have been able to get a clearer idea regarding what happens to achievement in the longer term”. 

A4: Thank you for this tip, we used the sentence as you proposed.

Minor comments:

5) The manuscript includes grammatical and syntax corrections. Please, provide a final version.

A5. We are sorry for the confusion, this time we tried to make it better. 

6) Page 2, Line 4: the acronym/abbreviation T1 has to be explained. 

A6. We did this, thank you. 

7) Page 2, Line 16: keyword students without special educational needs to be replaced by students without special educational needs and disabilities 

A7: Thank you for spotting this, however according to the second reviewer’s comment we changed the phrase into: students without disabilities.

8) Page 4, 11: needs a different word than ambiguous. I would suggest: “Inclusion in education is subject to many interpretations” 

A8: Thank you, we followed your suggestion. 

9) Page 5, Line 20: Add a short explanation (two extra lines maximum with a citation) of the neoliberal model of education.

A9: Thank you. We prepared a short explanation with a citation, as you wished.

10) Page 6, Line 14: How many of these 47 studies concerned students from primary and secondary education? 

A10: All the studies were conducted in primary and secondary education – we added this information. 

Please clarify.

11) Page 10, Line 18: Citation 53, add more recent citation to reinforce the argument such as:

a. Sirkko, R., Takala, M., & Wickman, K. (2018). Co-teaching in northern rural Finnish schools. Education in the North, 25(1–2), 217–238.

b. Scruggs, T. E., & Mastropieri, M. A. (2017). Making inclusion work with co-teaching. Teaching Exceptional Children, 49(4), 284–293. 

A11: Thank you for these suggestions – we have added them. 

12) Page 11, Line 5: Please mention the specific special education reforms in parenthesis. 

A12: We added some specific information about the years in which the reforms were implemented; however, in Poland those reforms do not have specific numbers or names. We added a reference - a manuscript about Polish educational systems in English, where it is possible to find more information. 

13) Page 11, Lines 8-9: “Parents of children with SEND … rather than special schools”: this is a strong statement please provide readers with a citation. 

A13: You are right that this statement was strong. We changed it to the information that parents can choose between regular and special classrooms and we added a reference.

14) Page 11, Lines 11-20: Please refer to the law/reform that describes this framework. 

A14: We added references to two works in which these issues were described in detail. 

15) Page 11, Lines 23-25: Please provide readers with citation of the law which justifies the higher number of co-teaching classes. 

A15: Thank you for turning our attention to this issue. This situation is not a result of law regulations, but it is connected with different factors, such as higher number of special education teachers in the cities, better job opportunities for them etc. 

Thank you very much for all your valuable comments and suggestions. 

Reviewer #2: Thank you for the opportunity to review the manuscript, “Academic achievement of students without special educational needs and disabilities in inclusive education - does the type of inclusion matter?”. The field is in need of more research on this topic, and I think this paper will be an important addition to the literature. Below I list my concerns with the current version of the manuscript:

1) Abstract-Specify that the study was conducted in Poland.

A1: We added this information. 

2) Introduction- Please consider reorganizing your literature review to create a more cohesive flow. There are some short paragraphs that could be combined and long ones that can be split. Also, I suggest moving the paragraph on page 3 line 21 to the current study subsection. Similarly, the first paragraph on page 6 and the second paragraph on page 7 should be moved to the current study subsection for a better flow. 

A2: We made thorough changes in the whole literature review section. We discussed your suggestions and we tried to find a solution, which is halfway between our primary solution and your suggestions. We hope that now the flow of the introduction is better. 

3) Additionally, I think the authors need to explain Poland’s educational system in terms of the educational stages and grades (could be in the inclusive education in Poland subsection). For example, on page 12 authors stated that they selected first-grade students. What is the grade equivalent of this in terms of PreK to 12? 

A3: Thank you very much for this suggestion. We understand that for readers who are not familiar with the Polish system the description could be unclear. Therefore, we described it in more detail in the ‘participants’ section. Our participants attended first grade lower secondary school (it was the 7th year of their compulsory education). 

4) Page 9 line 25 please explain what ad hoc interventions are. The reader needs examples to understand what this means. 

A4: Thank you very much for this comment. We added some examples: „Special education teachers or TAs in inclusive classrooms are often responsible for undertaking ad hoc interventions with students displaying disruptive behaviours, by, for example, asking them to calm down or leaving the classroom with the student to talk to him/her (…)” 

5) Method- consider adding a demographics table to the participant’s subsection. 

A5. We prepared a table according to your suggestion (Tabel 1).

6) Also, the authors stated that there were 174 students with disabilities in inclusive classrooms. It would be interesting to see their academic growth trajectories. If this was not planned as another paper, please consider broadening the aim of the paper for also students with disabilities and add a separate analysis. 

A6: Thank you for this suggestion. Unfortunately, students with disabilities learn only in two of three educational settings we described, so their results are not in line with this text. Therefore we decided to publish those results in a separate article. 

7) Results- no suggestion

8) Discussion- The authors said the study had three main aims, but the first aim was not mentioned in the paper earlier. I suggest putting this information in the introduction and rewriting the research questions. 

A7: Thank you very much for this comment. We added an additional section in the introduction on this topic. The analysis concerning this issue was moved to the results section. 

9) Page 20 line 5 “Our study has shown that in the Polish educational system there is informal tracking of…..” I am unconvinced that the study finds informal tracking. Please explain more. 

10) Page 20 line 18 Authors said, “In summary, the results of our study indicate that not only many teachers, but also many well-educated parents perceive inclusive classrooms as a risk factor for the development of students without SEND.” Please explain how your study indicates this conclusion. 

A8 & A9: Because we did not assess parents’ motivation for the choosing type of the classroom, we cannot be sure of our conclusions. Therefore, we used less strong words. However, in our opinion, if in traditional classrooms there is higher family SES compared to other types of classrooms it means that parents with high SES less often choose inclusive classrooms than traditional ones. 

11) Terminology- It’s not clear why the authors use students without special educational needs and disabilities instead of only students without disabilities. 

A10: In accordance with your suggestion we used the phrase ‘students without disabilities’ throughout the text. 

12) I believe the authors should revise the overall manuscript to improve the readability of the text.

A11: We made significant changes through the text.

Some minor issues

• Page 7 line 23 “This diverse solutions…..” consider revising the sentence 

• Page 8 line 22 “For instance, ……..” consider revising the sentence 

• Page 9 line 13 “Most studies show that……” please provide a reference 

• Page 9 line 20 “The positive effect” consider starting a new paragraph 

A12: All the mentioned issues were revised and changed. Thank you very much for your time to give us feedback – we really appreciate that.

---

## [Decision Letter · Decision Letter 1]

6 Jun 2022

Academic achievement of students without special educational needs and disabilities in inclusive education - does the type of inclusion matter?

PONE-D-21-30610R1

Dear Dr. Szumski,

We’re pleased to inform you that your manuscript has been judged scientifically suitable for publication and will be formally accepted for publication once it meets all outstanding technical requirements.

Kind regards,

Simona Lorena Comi

Academic Editor

PLOS ONE

Additional Editor Comments (optional):

Reviewers' comments:

Reviewer's Responses to Questions

**Comments to the Author**

1. If the authors have adequately addressed your comments raised in a previous round of review and you feel that this manuscript is now acceptable for publication, you may indicate that here to bypass the “Comments to the Author” section, enter your conflict of interest statement in the “Confidential to Editor” section, and submit your "Accept" recommendation.

Reviewer #1: All comments have been addressed

Reviewer #2: All comments have been addressed

2. Is the manuscript technically sound, and do the data support the conclusions?

Reviewer #1: Yes

Reviewer #2: Yes

3. Has the statistical analysis been performed appropriately and rigorously? 

Reviewer #1: Yes

Reviewer #2: Yes

4. Have the authors made all data underlying the findings in their manuscript fully available?

Reviewer #1: Yes

Reviewer #2: Yes

5. Is the manuscript presented in an intelligible fashion and written in standard English?

Reviewer #1: Yes

Reviewer #2: Yes

6. Review Comments to the Author

Reviewer #1: I really appreciated the thoroughness with which you delat my coments and accepted my suggestions. For the future it would be nice to move on with your research and try to implement this approach in other countries and conduct a cross-national research or even a longitudinal one.

Reviewer #2: Authors addressed all comments and the resulting manuscript would be a good addition to PLOS ONE.

7. PLOS authors have the option to publish the peer review history of their article (what does this mean?). If published, this will include your full peer review and any attached files.

Reviewer #1: **Yes: **Sofia Mastrokoukou

Reviewer #2: No

---

## [Editor Report · Acceptance letter]

9 Jun 2022

PONE-D-21-30610R1 

Academic achievement of students without special educational needs and disabilities in inclusive education – does the type of inclusion matter? 

Dear Dr. Szumski:

I'm pleased to inform you that your manuscript has been deemed suitable for publication in PLOS ONE. Congratulations! Your manuscript is now with our production department. 

Kind regards, 

on behalf of

Professor Simona Lorena Comi 

Academic Editor

PLOS ONE